# Poly (Lactic Acid)/Ground Tire Rubber Blends Using Peroxide Vulcanization

**DOI:** 10.3390/polym13091496

**Published:** 2021-05-06

**Authors:** Nicolas Candau, Oguzhan Oguz, Noel León Albiter, Gero Förster, Maria Lluïsa Maspoch

**Affiliations:** 1Centre Català del Plàstic (CCP), Polytechnic University of Catalunya, Barcelona Tech (EEBE-UPC), Av. D’Eduard Maristany 16, 08019 Barcelona, Spain; noel.leon@upc.edu (N.L.A.); gero-foerster@web.de (G.F.); maria.lluisa.maspoch@upc.edu (M.L.M.); 2Faculty of Engineering and Natural Sciences, Materials Science and Nano Engineering, Sabanci University, 34956 Istanbul, Turkey; oguzhanoguz@alumni.sabanciuniv.edu; 3Integrated Manufacturing Technologies Research and Application Center & Composite Technologies Center of Excellence, Sabanci University, 34906 Istanbul, Turkey

**Keywords:** poly (lactic acid) (PLA), wastes rubber, recycling, tensile properties

## Abstract

Poly (Lactic Acid) (PLA)/Ground Tire Rubber (GTR) blends using Dicumyl peroxide (DCP) as a crosslinking agent were prepared with the following aims: propose a new route to recycle wastes rubber from the automotive industry and improve the toughness and impact strength of the inherently brittle bio-based PLA. The GTR were subjected to two types of grinding process (cryo- and dry ambient grinding). Swelling measurements revealed the grinding to be associated with a mechanical damage of the rubber chains, independently on the type of grinding or on the GTR size (from <400 µm to <63 µm). Moreover, the finest GTR contains the largest amount of reinforcing elements (carbon black, clay) that can be advantageously used in PLA/GTR blends. Indeed, the use of the finest cryo-grinded GTR in the presence of DCP showed the least decrease of the tensile strength (−30%); maintenance of the tensile modulus and the largest improvement of the strain at break (+80%), energy at break (+60%) and impact strength (+90%) as compared to the neat PLA. The results were attributed to the good dispersion of both fine GTR and clay particles into the PLA matrix. Moreover, a possible re-crosslinking of the GTR particles and/or co-crosslinking at PLA/GTR interface in presence of DCP is expected to contribute to such improved ductility and impact strength.

## 1. Introduction

Due to the generation of mega-tons of plastics and rubber wastes each year in Europe, the polymer industry is facing a considerable ecological risk [1]. To tackle this issue, the recycling of wastes rubber [2,3] and the elaboration of bio-based plastics [4,5] have shown a wide development within the last decade. By using these ecological strategies, thermoplastic elastomers (TPE) [6,7,8,9,10] using conventional thermoplastics from crude oil such as polyethylene (PE), polypropylene (PP) or polyethylene terephthalate (PET) and fresh elastomers such as ethylene propylene diene monomer (EPDM), natural rubber (NR) or styrene butadiene rubber (SBR) were progressively replaced by polymeric blends containing bio-based thermoplastics [11,12,13], wastes thermoplastics [14,15] or wastes rubber [16,17]. Initially based on an ecological demand, the industrial production of these green materials also requires mechanical performances (stiffness, strength, ductility) comparable to the ones of conventional thermoplastic elastomers they intend to replace. 

The progressive replacement of fresh rubber by Ground Tire Rubber (GTR) in TPE aims to participate in the circular economy of the automotive industry. Nonetheless, these blends generally contain a limited quantity of wastes rubber as thermoplastic properties rapidly deteriorate at GTR amounts of 10–20 wt% of GTR [18,19]. This may be due to the presence of large rubber particles [20] or their facility to aggregate due to their weak miscibility and poor interfacial adhesion with the plastic phase [21]. The interfacial properties between GTR and the plastic matrix can however be drastically improved by reducing the GTR particle size [21,22], by using compatibilizers [22,23,24] and vulcanizing agents [24,25,26] or encapsulating wastes rubber into a fresh rubber phase [17,23].

Among the recently developed renewable bio-based polymers, Poly(lactide) (PLA) is one of the most promising. Its high tensile strength and stiffness compete with those of petroleum-based commodity plastics (PP, PE, PET). Due to its biodegradable, bio-compatible and compostable nature, the market demand for PLA based composites has dramatically grown over the past decade, with a primary use for single use packaging applications. However, the inherent disadvantages of PLA are its intrinsic brittleness and poor ductility impede its use for applications such as fiber manufacture, automotive industry or building construction [27]. Therefore, it is necessary to improve the ductility of PLA to extend its applications where toughness and impact resistance are crucial. 

To improve its toughness and impact resistance, PLA has been blended with synthetic fibers [28], natural fibers [29,30], toughening polymers [31,32], in presence of reactive compatibilizers [33,34] or using dynamic vulcanization [35,36]. Among the additives, compatibilizers or toughening polymers cited above, elastomers have been shown to exhibit among the most interesting toughening effects [35,36,37]. However, they are not cost-effective and can hardly compensate the relatively high cost of PLA which compares that of common commodity plastics. Hence, it is still a challenge to investigate more cost-effective PLA impact elastomeric modifiers [38]. The use of waste rubber materials from the pneumatic industry is seen as a possibility. In addition, the replacement of fresh elastomers (e.g., natural rubber) by wastes tire rubber responds to a growing demand for a circular economy of the pneumatic industry.

The degradability of PLA by main chain scission [39] and of wastes rubber by reversion of polysulfide bonds [40,41] make the preparation of PLA/GTR composites challenging [42,43,44,45]. Scrap rubber from tires after thermal shock method [42] and frost shattering method [43] were blended with PLA. Tensile and impact properties of PLA were found to be lowered by wastes rubber addition due to poor adhesion between the wastes rubber and PLA matrix. However, the use of silane agent as compatibilizer in PLA/GTR blends showed increased strain at fail and impact strength while elastic modulus and strength decreased moderately for an optimal GTR content of 15 wt% [44,45]. Finally, the use of a vulcanizing agent had been demonstrated to show significant improvement of mechanical properties of PLA/Natural Rubber (NR) blends [46,47,48] but has not been investigated yet in PLA/GTR blends.

This study presents an investigation dedicated to the preparation of PLA-GTR blends using dicumyl peroxide (DCP) as a crosslinking agent. A systematic study has been conducted regarding the role of GTR particle size, the type of GTR grinding process, the DCP influence and the quantity of wastes rubber introduced into the PLA/GTR blends. In the first section, the properties of the GTR particles were studied. Their composition, size distribution and crosslink density were discussed which allowed to make a careful selection of the GTR particles to be further blended with PLA. In the second and third sections, the tensile and impact properties of the PLA-GTR blends were presented. It has been found that the addition of 15 wt% of the finest cryo-grinded GTR in presence of DCP showed the least decrease of the strength, a maintain of the elastic modulus, the most improved ductility, energy at break and impact strength as compared to the neat PLA. The prepared PLA/GTR based composites may find future applications in structural parts by offering cost-effective alternative to conventional commodity plastics (PE, PP, PET). Especially, they may be used as structures or semi-structures in the automotive industry (interior automotive parts), building construction (3D-printed parts) or consumer goods (cell phone casings). 

## 2. Materials and Methods

### 2.1. Materials Composition 

The PLA2002D^®^ extrusion grade was obtained from NatureWorks (Arendonk, Belgium). Its number-average molecular weight, Mn = 99 kg.mol^−1^ and weight-average molecular weight, Mw = 187 kg.mol^−1^ have been determined using Size Exclusion Chromatography (SEC) [49]. Ground tire rubber (GTR) was supplied by the company J. Allcock & Sons Ltd. (Manchester, United Kingdom) using the transformation of tire buffing into finer rubber crumbs via a controlled dry-ambient grinding (GTR_a_) or cryo-grinding (GTR_c_). The obtained GTR contains rubber and carbon black (CB). The rubber is composed of Natural Rubber (NR) and Styrene Butadiene Rubber (SBR). The GTR particles were subsequently separated into different sizes using a vibratory sieve shaker (Analysette 3). The sieving was performed during 5 min in dry conditions with a vibratory amplitude of 0.15 mm. The following sizes were extracted for the GTR_a_: 40 ’s mesh (size <420 µm), 80 ’s mesh (size <180 µm) and 120 ’s mesh (size <125 µm). For the cryo-grinded GTR_c_, the following sizes were extracted: 120 ’s mesh (size <125 µm) and 230 ’s mesh (size <63 µm). The size distribution of the GTR particles was determined using an ImageJ treatment of optical microscopy images of the GTR particles. Before melt-blending, the PLA was dried overnight in a vacuum oven (Vaciotem-TV, J.P. SELECTA^®^ Barcelona, Spain) to prevent humidity absorption, over silica gel at 70 °C to remove any moisture. The sieved GTR crumbs were dried under the same conditions. 

### 2.2. Materials Processing

Melt blending was performed in an internal mixer (Brabender Plastic-Corder W50EHT, Brabender GmbH & Co. Duisburg, Germany) using two counter-rotating screws (roller blade type “W”). After optimization of the processing conditions, the processing temperature was chosen equal to 170 °C and the rotation speed equal to 60 RPM. The PLA was first added, and an antioxidant (Irganox^®^ 1010, BASF, Ludwigshafen, Germany) was used (0.2 wt% of the total weight of the PLA/GTR blend) to prevent PLA degradation during the blending. After 5 min (stabilization of the torque), the GTR rubber crumbs were added. 5 min after the introduction of GTR (stabilization of the torque), the dicumyl peroxide (DCP) is finally added (1.5 g per hundred grams of GTR) as vulcanizing agent. The blends were then hot-pressed at 1 MPA and 170 °C during 5 min in a LAP PL-15 plate press (IQAP Masterbatch SL) using a mask of 1 mm thickness. The plate was subsequently cooled down to room temperature with a cooling rate of 50 °C·min^−1^. Such fast cooling was chosen so that the obtained PLA/GTR do not re-crystallize (the DSC crystallinity was measured below 2 wt% for all prepared blends). The nomenclature of the different blends is presented in Table 1.

### 2.3. GTR Size Distribution

The images of the GTR particles were taken using a polarized light microscope (Optekusa, Zoom Stereo Microscope SCZ-T4P, Scotts Valley, CA, USA). The rubber particles were put on a glass plate and the particles were then dispersed by immersing into water. An open-source software (ImageJ version 1.51) was used for image analysis. The GTR particles were assumed to adopt an elliptical shape, and the semi-major axis *a* and the semi-minor axis *b* were calculated. The mean diameter was calculated using the formula *d = 2(2a + b)/3*.

### 2.4. Scanning Electron Microscopy (SEM)

The fracture surfaces of the specimens were observed after tensile testing with a field emission scanning electron microscope (JSL-7001F, JEOL, Tokyo, Japan). A few nanometers thick conductive layer of a Pt_80_/Pd_20_ alloy was sputtered on the fracture surface using a high-resolution sputter coater (Cressington 208HR) in order to avoid electron charging on the specimen surface. The surface topography was observed with a voltage of 1 kV. Chemical analysis was performed by EDX with a voltage of 20 kV. 

### 2.5. Thermogravimetric Analysis (TGA)

Thermogravimetric analysis is performed on GTR particles using a STAR^e^ system (Mettler Toledo, Columbus, OH, USA). The particles are put into alumina crucible with a quantity around 5–10 mg. The standard IEC 60811-100 is used for the determination of the carbon black content. To do so, the GTR are heated from 30 to 1000 °C with a heating ramp of 10 °C/min working under nitrogen environment from 30 to 850 °C, and under air environment from 850 to 1000 °C.

### 2.6. Swelling

GTR is immersed in cyclohexane for 72 h, and the solvent is changed every 24 h. After 72 h, the swollen mass of (m_s_) is measured. The GTR are then placed in an oven under vacuum at 70 °C during 6 h to remove the solvent. The mass of the dry samples (m_d_) is then measured. The swelling ratio of the specimen *Q* is calculated. The network chain density is calculated from swelling and the Flory–Rehner equation: (1)$$\upsilon =\frac{ln\left(1-v_{2}\right)+v_{2}+\chi_{1}v_{2}{}^{2}}{V_{1}\left(-v_{2}{}^{\frac{1}{3}}+\frac{2}{f}v_{2}\right)}$$

With *v*_2_ = 1/*Q*_B_. *V**_1_* = 108 cm^3^/mol^−1^ is the molar volume of the solvent (cyclohexane), c*_1_* is the Flory–Huggins polymer solvent dimensionless interaction term (c*_1_* is equal to 0.353 for the GTR-cyclohexane system). The ratio *2/f* is associated with the phantom model that assumes spatial fluctuation of crosslinks (non-affine) used for high deformation ratios. *f*, the crosslink functionality, is chosen equal to 4. GTR contains non-rubber particles like carbon black. Hence, the Kraus correction [50] is used to account for the contribution of filler in swelling ratio, assuming that they do not contribute to swelling. *Q**c* is the swelling ratio of the rubber matrix defined as follows: (2)$$Q_{c}=\frac{Q-\varphi }{1-\varphi }$$
with *ϕ* is the volume fraction of fillers. Krauss correction in Equation (2) assumes non-adhesion of the fillers to the rubbery matrix in the swollen state.

### 2.7. Thermoporosimetry

GTR is put into cyclohexane during 72 h to reach the swelling equilibrium. They are then carefully extracted and put into an aluminum crucible. A Q2000 DSC (TA Instruments) is used. The sample is first cooled down to −50 °C at 10 °C/min followed by an isothermal step at −50 °C during 2 min. The sample is then heated at 10 °C·min^−1^ up to 30 °C, during which endothermic peaks correspond to melting *T_m_* of the cyclohexane entrapped in the network. Melting peaks are deconvoluted, and the intensity is normalized by the swollen weight. The full procedure developed for vulcanized natural rubber [51,52] and EPDM [53,54] is used here for rubber particle wastes. By derivation of the Gibbs-Thompson equation, the normalized pore size is given by
(3)$$\frac{L}{L_{f}}=\frac{T_{m}{}^{0}-T_{f}}{T_{m}{}^{0}-T}$$
where *T_f_* and *L_f_* correspond to the melting temperature and the size of the largest pores entrapped in the network, respectively. After derivation of Equation (3), the normalized intensity distribution of the pore size is given by
(4)$$I_{n}=\frac{1}{m}\frac{dH}{dT}\frac{{\left(T_{m}{}^{0}-T\right)}^{2}}{L_{f}}$$

*L_f_* value being unknown, the intensity *I = L_f_I_n_* is hence plotted instead *I_n_* to account for the pore size distribution. The average normalized pore size is then calculated as the *L/L_f_* value associated with half of the area under the normalized signal.

### 2.8. Uniaxial Tensile Stretching (UTS)

Dogbone shaped specimens of type 1BA were extracted from hot molded sheets by die-cutting with a specimen preparation punching machine (CEAST) shortly after the hot molding process. The specimens were then stored at room temperature for one week prior to testing in order to provide realistic industrial storage conditions of the processed material. Uniaxial tensile tests according to the ISO 527 standard were performed on a universal testing machine (SUN 2500, GALDABINI, Cardano al Campo, Italia) at room temperature and a constant crosshead speed of 10 mm/min. The machine was equipped with a video extensometer (OS-65D CCD, Minstron, Taipei, Taiwan). Tensile modulus is measured in the linear regime up the deformation of 1%. The tensile strength is calculated as the maximum stress reached after the elastic regime and directly read from the engineering strain–stress curve. The strain at break is measured by direct read from the engineering strain–stress curve, and the energy at break is calculated as the area under the engineering stress–strain curve until failure.

### 2.9. Impact-Tensile Tests

Impact tensile tests have been performed using a swinging pendulum (CEAST 6545, Torino, Italy) having a length L = 374 mm, assembled with a hammer having a mass of 3.655 kg and a potential energy of 25 J, is released from an angle of 45° and hits the specimen at its lower position with an impact energy of 3.93 J and an impact velocity of 1.47 m/s. The specimen, clamped with a crosshead of 60 g, is submitted to a high-speed tensile load. The tensile-impact strength $a_{tU}$, defined as the energy absorbed by the specimen until the fracture divided by the initial cross section $A_{0}=t*b$, with t the thickness and b the width of the sample, has been determined through tensile-impact testing according to the standard ISO 8256. The same type of specimens as for tensile testing has been used (type 1BA), and the tests have been performed two weeks after the processing of the sheets.

## 3. Results

### 3.1. Ground Tire Rubber (GTR) Properties

Prior to the mechanical characterization of the PLA/GTR blends, the properties of the GTR particles were studied. At a macro-scale, their size distribution, degradation properties and chemical composition (Figure 1 and Figure 2) were estimated. At the chains network scale, the average density and the distribution of the chains network depending on the applied grinding process and sieving mesh sizes (Figure 3) were determined.

The GTR particle size distribution was drastically reduced after sieving with the largest mesh (Figure 1a). As cryo-grinding usually results in smaller particle size as compared to ambient grinding [55], it was possible to sieve sufficient quantity of cryo-grinded GTR at higher mesh 230 ’s (size <63 µm). The thermal stability of the sieved crumb GTR obtained from ambient and cryo-grinding processes is discussed based on TGA curves (Figure 1b,c). For all GTR, the first derivative peak around 360–380 °C is ascribed to the degradation of Natural Rubber, the second one situated around 420–445 °C to the degradation of SBR [56]. The remaining mass above 500 °C is mostly ascribed to non-rubber components, essentially the carbon black (CB) particles. The degradation temperature of NR and SBR seems to not drastically depend on the nature of the grinding or sieving. However, the fraction of non-rubber components is found to largely increase by decreasing the GTR size. This is possibly due to a gravity effect arising from the sieving process, as the fine and heavy non-rubber components, free or attached to the finest GTR particles, preferentially go through the sieves.

Grinding processes generally require clay minerals. Their presence is indeed shown by the preponderance of Magnesium (Mg) atoms as observed by EDX for highly sieved GTR_c_ (Figure 2). The sulfur arising from the vulcanization (curing) of the tire is also present in the GTR. The mass ratio of Magnesium (Mg) over Sulphur (S), Mg/S, obtained from the quantitative EDX analysis of chemical elements is found equal to 0.13 and 2.26 for GTR_a_ 120’s mesh and GTR_c_ 120’s mesh, respectively (Figure 2a,b), suggesting the presence of clay particles to be negligible in sieved ambient grinded GTR_a_ as compared to cryo-grinded GTR_c_. This quantitative EDX study also reveals that the clay particles are more present into sieved cryo-grinded particles obtained with highest sieving mesh (Mg/S ratio is of 3.67 against 2.26 for GTR_c_ 230’s mesh and GTR_c_ 120’s mesh, respectively, Figure 2b,c). The presence of these non-rubber components may explain the small TGA derivative peak close to 600 °C (Figure 1c) observed for the finest GTR_c_ particles. At such high temperature, minerals like clay [57] or Kaolin [58] indeed start to decompose. One may note however that, while most of the rubber decomposes below 500 °C, it has been shown that the formation of short hydrocarbons attest for the thermal degradation of the rubber chains above 500 °C [59]. In the following, the estimation of the network chain density of the GTR will be corrected from the presence of these non-rubber components, namely, carbon black particles and clay minerals (Figure 3).

The ground tire rubber crumbs produced by dry ambient and cryo-grinding of the used tire buffing were swollen into cyclohexane and the average network chain density and its distribution analyzed (Figure 3a–c). GTR showed an overall increase of the swelling ratio, *Q*, from un-grinded (Q ~ 1/0.45) to grinded GTRa (Q ~ 1/0.25) and GTRc (Q ~ 1/0.35). The lower swelling ratio of GTR_c_ as compared to GTR_a_ is likely explained by an increasing amount of non-rubber components in GTR_c_, mostly carbon black (CB), as suggested by the TGA (Figure 1b,c). After correction form the presence of non-rubber components, and assuming no adhesion between these rigid particles and the swollen rubber, the rubber network chains density has been calculated from the swelling ratio and using the Flory–Rehner equation (Equations (1) and (2)). The network chains density was found close to 1 × 10^−4^ mol.cm^−^³ for all grinded GTR, independently on the size and on the grinding type (Figure 3b). These values are found much lower that the network chain density of the ungrinded GTR, found around 5.9 × 10^−4^ mol.cm^−3^. This suggests the different tested grinding processes to show similar ability to damage the chains network and that the finest GTR size does not result from more intense damage. At molecular scale, such damage may traduce the rubber chains scission, sulphur-bonds breakage or rubber-filler rupture (Figure 3b). Possibly, the filler aggregates present into the GTR may also undergo filler–filler rupture, as had been discussed in the case of mechanically damaged carbon black filled rubbers [60,61,62].

Thermoporosimetry experiment is further used to reveal the distribution of the distance between the nodes of the rubber chains network (crosslinks or entanglements) of the GTR (Figure 3c). This method has been widely used in the case of bulk vulcanized rubber [63,64]. It is applied here in the case of ground tire rubber particles. Thermoporosimetry is based on the quantification of the distribution of the melting temperature of a crystallized rubber solvent trapped into the rubber network. Then, through the use of the Gibbs–Thomson equation (Equations (3) and (4)), the distribution of the crystallite sizes is calculated. Assuming these crystallites to be constrained by the network nodes (trapped physical entanglements or chemical crosslink), the distribution of the melting temperature directly relates to the distribution of the distance between the nodes (crosslinks or trapped entanglements). This distance is defined as the pores size. As indicated in Figure 3c, the shift of the average pore size to higher values as well as the increased distribution from un-grinded to grinded GTR is the result of more intense but heterogeneous damage of the chains network due to the mechanical cutting. Consistent with measures of the network chain density (Figure 3a,b), the pore size distribution is found very similar in all grinded GTR. In the following, it will be admitted that all sieved GTR exhibit similar average network chains densities with rather broad distribution. Such properties are found independent on the size and grinding type. The level of damage in the network chain density of the GTR as well as their non-rubber content are expected to have both a role in the tensile behavior of the PLA/GTR blends, as will be discussed in the subsequent sections. 

### 3.2. Tensile Properties of PLA/GTR Blends: Effect of the GTR Size

PLA/GTRa and PLA/GTRc blends using 15 wt% of sieved dry ambient grinded GTRa and sieved cryo-grinded GTRc were processed, and their tensile properties presented (Figure 4 and Figure 5). While the swelling methods cannot indicate the percentage of chains scission or sulphur bond breakage occurring during grinding, the low network chain density of the GTR (Figure 3b), resulting from the mechanical cutting is expected to, at least partially, result from devulcanization (sulphur bond breakage). GTR treated by grinding is indeed expected to possess a certain reactivity, and a possible re-vulcanization can be envisaged [65]. To this aim, the effect of a curing agent (DCP) has been investigated in the case of PLA/GTRa blends. One may note that the effect of DCP has also been investigated for PLA/GTRa blends, on a selection of the finest GTRc particles and for a unique GTRc content of 15 wt% (see Section 3.3).

Stress–strain tensile curve of neat PLA shows linearity up to 2% of deformation followed by a yielding at around 3%. No post-yielding deformation is noted as the PLA rapidly breaks at 3.5% (Figure 4). This brittle behavior of PLA is due to the storage at ambient temperature sufficiently long to cause a ductile to brittle transition due to physical aging [66]. The addition of 15 wt% of GTR results in a decreased yield strength, arising from rubber particles softening and likely voids formation at PLA/GTR interface. However, a plastic plateau is observed, indicating the PLA/GTR blends to be more ductile as compared to neat PLA.

In all PLA/GTR blends, the stiffness and strength are found lower that the ones of the neat PLA, expectedly, due to the introduction of rubbery particles into the glassy PLA matrix. The elastic modulus and yield strength in PLA/GTR_a_ seem weakly dependent on the particle sizes (Figure 5a,b). However, the elastic modulus of the neat PLA is recovered for the PLA/GTR_c_ blends incorporating the finest GTR_c_. The strength is also increased consistent with previously reported results in TPE using wastes rubber in the same range of sizes [24]. This notable rise in stiffness and strength with lowering the GTR sizes may additionally find origin in the presence of significant non-rubber fraction in the finest GTR_c_ (more than 50 wt% from TGA, Figure 1b), mostly composed by carbon black particles that contribute to mechanically reinforce the GTR and by inference the PLA/GTR blend. 

Moreover, the micron sized clay particles used in cryo-grinding process (Figure 2b,c) are found to be dispersed into the PLA during melt-blend with GTR, as shown by the fracture surface of the PLA/GTR_a_ 15 wt% sieved 230’s mesh (Figure 6a–c). In particular, the EDX images show the mapping of magnesium (Mg) indicative of the location of the micron-sized clay particles (Figure 6b), while the mapping of the sulphur (S) is indicative of the location of the GTR particles (Figure 6c). It is worth to note that the clay particles are more likely distributed in the PLA matrix (detached from the GTR particles). The inclusion of micron sized clay particles into the PLA matrix is expected to contribute to the mechanical reinforcement of PLA (increased strength and stiffness) as it has been demonstrated in the literature [67,68], hence contributing to the recovery of elastic modulus and partial recovery of strength of the neat PLA in the PLA/GTR_a_ sieved 230’s mesh.

The strain and energy at break both show visible increase when incorporating finer GTR, independently on the type of grinding (Figure 5c,d). A homogeneous particle distribution into the PLA matrix is suggested by SEM images (Figure 11 and Figure A1). Hence, the wider number of distributed and fine GTR particles distributes the stress upon loading and may result in cavitation/decohesion at PLA/GTR rather than in brittle failure through the development of crazes as usually the case in the neat brittle PLA [22,24].

The addition of the vulcanizing agent, DCP, is found to result in an overall increase of the strain and energy at break. While the effect may be somehow hindered by the error bar, it is however more pronounced in blends using the finest GTR particles. DCP had been demonstrated to be an efficient crosslinking agent for PLA and natural rubber (NR) [12,13,48]. The presence of DCP to form free radicals may crosslink both PLA and GTR particles, as they are assumed to be partially devulcanized regarding their low network chains density (Figure 3b). The action of free radicals at PLA/GTR interface is expected to be more efficient for the finest GTR as their containing more potential crosslinking sites due to their large surface area. 

### 3.3. Tensile Properties of PLA-GTR Blends: Effect of the GTR Content 

The effect of the GTR content from 0 to 30 wt% has been studied using the PLA/GTR_a_ blends containing the finest GTR_a_ 120’s mesh (Figure 7 and Figure 8). The increasing amount of GTR results in a decreased strength, arising from rubber particles softening. For all rubber content and in presence of crosslinking agent or not, stress–strain tensile curves of the PLA-GTR_a_ blends (Figure 7) show more ductile behavior with a larger plastic plateau as compared to the brittle behavior of neat PLA, as previously discussed (Figure 4). 

The increasing amount of rubber particles results in a drop in tensile modulus, *E_T_*, and tensile strength, *UTS,* of the PLA/GTR_a_ blends (Figure 8a,b). The introduction of DCP has beneficial effect on the elastic modulus. This effect is more pronounced with an increased amount of GTR and may results in (i) crosslinking of the PLA matrix as suggested in Ref. [69], (ii) re-crosslinking of the GTR, (iii) co-crosslinking at PLA/GTR interface that all potentially participate in the increased stiffness of the PLA/GTR blend. 

By increasing the GTR content from 0 to 22.5 wt%, the strain and energy at break increase to reach a maximum at an optimum quantity of GTR between 7.5 and 15 wt% (Figure 8c,d). However, a drastic decrease of strain and energy at break above this optimum may be caused by an agglomeration of the rubber particles. The DCP shows slight increase in the strain and energy at break in the PLA/GTR_a_ blends for a content at and above 15 wt%. 

The effect of the GTR particle content, from 0 to 30 wt% has been studied for the blends containing the finest GTR_c_ particles, namely PLA/GTR_c_ 230’s (Figure 9 and Figure 10). The addition of increasing amount of GTR results in a decreased strength but to a lower extent as compared to blends using GTR_a_ (Figure 7 and Figure 8). Stress–strain tensile curves of the PLA-GTR blends (Figure 9) show more ductile behavior with a larger plastic plateau as compared to the brittle behavior of neat PLA. 

Interestingly, the introduction of GTR particles at low amount (3 wt%) results in an increase of tensile modulus, *E_T_*, and only small drop in tensile strength of the PLA/GTR_c_ blends (Figure 10a,b). Above this content, a decrease of the elastic modulus and tensile strength is observed, but much lower as compared to PLA/GTR_a_ (Figure 8a,b). These results are likely explained by (i) the finer rubber particles extracted from cryo-grinding and (ii) the higher quantity of non-rubber reinforcing elements in blends using finest cryo-grinded GTR, sized clay (Figure 1a,b).

By increasing the GTR content up to 30 wt%, the strain and energy at break increase with the quantity of GTR introduced (Figure 10c,d). The maintain of higher values of strain and energy at break as compared to the one of neat PLA and to the PLA/ GTR_a_ blends (Figure 8c,d) is allowed by the fine GTR size, expected to be well dispersed into the PLA matrix (see SEM images on Figure 11). The influence of the crosslinking agent is evidenced here in PLA/GTR_c_ 15% and PLA/GTR_c_ 30% blends as it results in a mutual increase of the elastic modulus, strain at break and energy at break (Figure 10a–d). 

The fracture surface at cross-section of the mechanically tested PLA and PLA-GTR were analyzed by SEM (Figure 11 and Figure A1). The deformation mechanism of the brittle PLA had been demonstrated to result in the formation of surface crazes [70], resulting in a smooth fracture surface (Figure 11a,b). In PLA/GTR_a_ and PLA/GTR_c_ blends with GTR up to 15 wt%, the micrographs show a relatively homogeneous dispersion of the GTR particles in the PLA matrix (Figure 11c–h). This is confirmed by EDX images as attested by the sulphur-rich domains indicating the presence of the GTR particles (Figure A1). Moreover, SEM micrographs reveal a good adhesion of some of the GTR particles to the PLA matrix (Figure 11c–h) while some other show partial decohesion, resulting from damage occurring during tensile test. As previously commented, these damage mechanisms may reduce the stress concentration at PLA/GTR interface and result in a larger plastic deformation that in neat PLA (Figure 7 and Figure 9). However, their maintain up to the largest deformation possible should result in delayed macroscopic failure, which would be facilitated for by the presence of the DCP as crosslinking agent. 

The materials showing the most promising mechanical performance measured during tensile test, namely PLA/GTR_c_ with 230’s mesh sized GTR particles (Figure 9 and Figure 10), had then been subjected to tensile impact. Consistent with increased tensile energy at break of the PLA/GTR blends as compared to the more brittle PLA (Figure 10d), tensile impact strength also increases in the PLA/GTR blends (Figure 12). Moreover, a 1-fold increase of the impact strength of crosslinked PLA/GTR 15% is found as compared to un-crosslinked PLA/GTR 15%, highlighting the role of DCP as crosslinking agent to prevent macroscopic failure in impact conditions (high strain loading). Crosslinked PLA/GTR blends using 15% of fine cryo-grinded GTR comply with the requirements for a sustainable route for wastes rubber recycling. Moreover, they represent a reasonable compromise in terms of mechanical performance. In spite of a 30% drop of the PLA tensile strength, the elastic modulus of PLA is maintained, the strain at break is increased by 80%, the energy at break is increased by 60%, and the impact strength is increased by 90%.

In addition to their cost-effectiveness and interest for the circularity of the pneumatic industry, the use of wastes rubber as substitute of fresh rubber may find interest from an industrial point of view due to their promising mechanical performances. PLA had been blended with fresh rubber, such as synthetic isoprene rubber (IR) [71], nitrile butadiene rubber (NBR) [72,73], natural rubber (NR) [48,74] and epoxidized natural rubber (ENR) [75,76]. As compared to these composites, our PLA/GTR_c_ 15% is found to show a maintain of the PLA elastic modulus while, at similar rubber content, PLA/NR blends show a decrease of more than 40% [48,74]. The decrease in strength (−30%) of the PLA/GTR_c_ 15% is similar to those in unvulcanized PLA/NR and dynamically vulcanized PLA/NR blends [48,74]. Its 80% increase in strain at break as well as 90% of impact strength are comparable to those of dynamically vulcanized PLA/NR [48]. 

However, it has been shown that, for rubber content higher than 15 wt%, PLA/NR or PLA/SBR blends show much higher impact strength as compared to our materials. Hence, one possible perspective to the composites developed in the present study is the preparation of ternary blends using PLA, GTR and fresh rubber whose composition can be tuned to design toughened PLA based composites with maintained stiffness and strength.

## 4. Conclusions

Poly (Lactic Acid) (PLA)/Ground Tire Rubber (GTR) blends using crosslinking agent were prepared as a route to recycle wastes rubber from the automotive industry (GTR) and improve the toughness of the bio-based brittle PLA. Firstly, the physico-chemical properties of the GTR were investigated; secondly, the tensile and impact properties of the PLA/GTR blends were discussed.

It has been found that the grinding (mechanical cutting) of the GTR resulted in a wide mechanical damage of the rubber network as attested by the decrease of their chains network density which may result from sulphur-bonds breakage (devulcanization) but also chains scission. The GTR particles treated by grinding are hence expected to possess a certain reactivity favorable to a re-vulcanization. Moreover, it has been found that the finest sieved GTR particles were accompanied by the largest amount of non-rubber reinforcing components (carbon black particles, clay).

Based on the prior GTR characterization, PLA/GTR blends have been processed by using DCP. The use of the finest cryo-grinded GTR in the presence of DCP showed the least decrease of the tensile strength (−30%), maintenance of the tensile modulus and the largest improvement of the strain at break (+80%), energy at break (+60%) and impact strength (+90%) as compared to the neat PLA. The results were attributed to the following factors: the good dispersion of the fine GTR particles into the PLA matrix, the partial re-crosslinking of the GTR particles and co-crosslinking at PLA/GTR interface and the presence of reinforcing carbon black into the GTR particles and clay particles dispersed into the PLA matrix. The degree of grafting of the PLA chains on the GTR surface would be an interesting prospective investigation in order to bring more insight on the interfacial properties of such composite.

The promising tensile and impact properties suggest the obtained PLA/GTR blends to be categorized as toughened semi-degradable composites. However, rubber from tires, as thermoset materials, possess a crosslinking network whose degradation by a controlled devulcanization process is a challenging question to solve. In the future, the improvement of the degradability of these PLA/GTR composites may be investigated by using the conjunction action of degrading microorganisms for PLA [77] and wastes rubber [78]. 

## Figures and Tables

**Figure 1 polymers-13-01496-f001:**
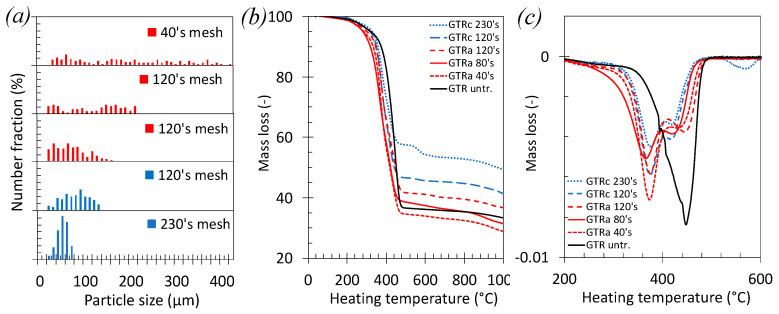
(**a**) Size distribution of dry ambient grinding and cryo-grinding of GTR particles, (**b**) mass loss versus heating temperature obtained from TGA measurements and (**c**) first derivative of the mass loss versus heating temperature.

**Figure 2 polymers-13-01496-f002:**
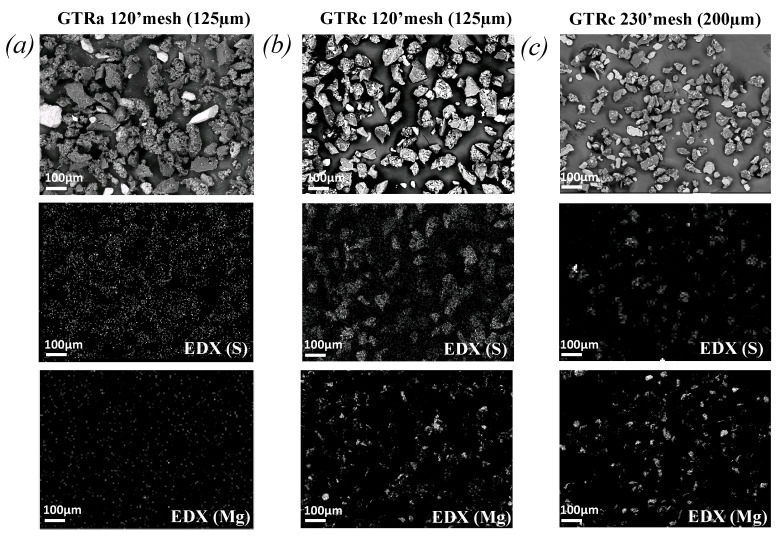
SEM images of GTR_a_ particles with 120’s mesh (**a**), GTR_c_ particles with 120’s mesh (**b**) and GTR_c_ particles with 230’s mesh sieving (**c**) at a magnitude 100× (top figures). Mapping of the sulphur contained into the GTR particles (center figures) and mapping of the Mg (bottom figures) obtained from chemical analysis by EDX.

**Figure 3 polymers-13-01496-f003:**
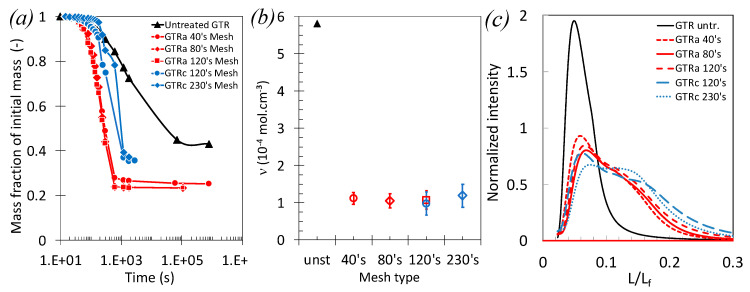
(**a**) Deswelling kinetics in swollen grinded GTR crumbs of different sizes displayed by mass fraction of the initial mass versus logarithmic time. (**b**) Network chains densities of grinded GTR crumbs of different particle sizes whose mesh types are detailed in the experimental section. (**c**) Normalized intensity versus normalized pores size of grinded GTR crumbs of different particle sizes extracted from thermoporosimetry experiments (see Experimental Section for more details on the procedure).

**Figure 4 polymers-13-01496-f004:**
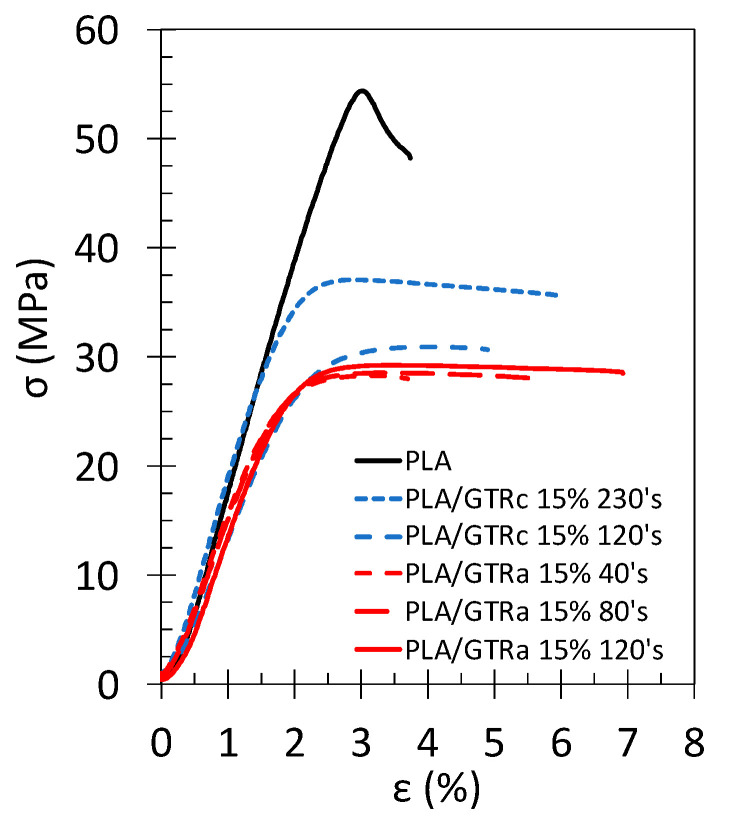
Engineering stress–strain curves of tensile test performed at 10 mm·min^−1^ and at room temperature on the neat PLA (black continuous line), PLA/GTR_a_ 15 wt% sieved 40’s mesh (red dashed dotted line), PLA/GTR_a_ 15 wt% sieved 80’s mesh (red dotted line), PLA/GTR_a_ 15 wt% sieved 120’s mesh (red continuous line), PLA/GTR_c_ 15 wt% sieved 120’s mesh (blue large dotted line) and PLA/GTR_a_ 15 wt% sieved 230’s mesh (blue small dotted line). Tensile curves shown in this study correspond to PLA/GTR blends using DCP at 1.5 wt% of the GTR. The effect of DCP content is further detailed in Figure 5.

**Figure 5 polymers-13-01496-f005:**
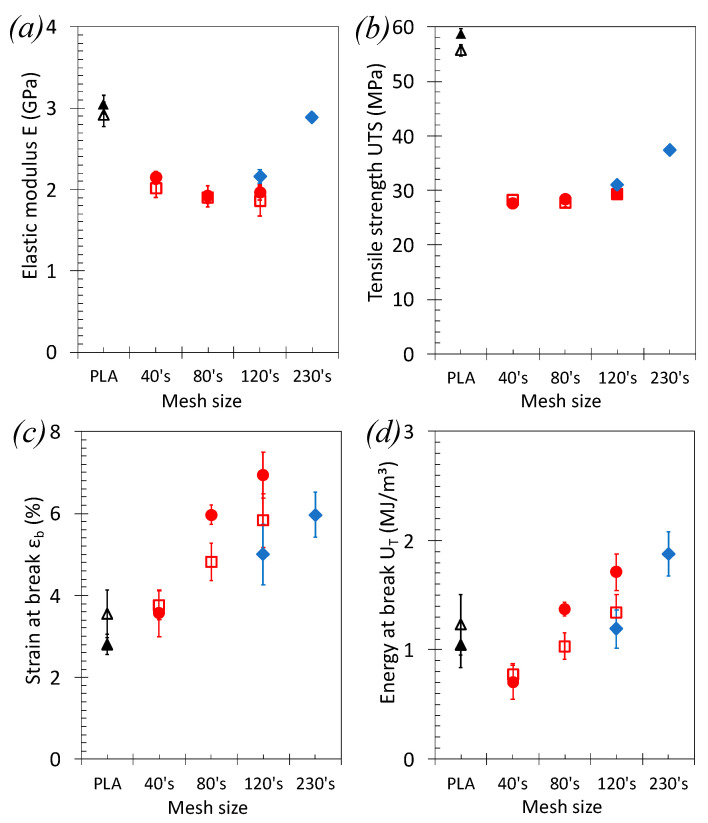
Effect of the GTR size on the tensile properties of PLA/GTR blends. (**a**) Elastic modulus, E_T_, (**b**) tensile strength UTS, (**c**) strain at break, ε_b_, and (**d**) energy at break, U_T_, measured from tensile test performed at 10 mm·min^−1^ and at room temperature: neat PLA in absence of DCP (black unfilled triangle symbols), neat PLA in presence of 1.5 wt% DCP (black filled triangle symbols), PLA/GTR_a_ 15% in absence of DCP (red square symbols), PLA/GTR_a_ 15% in presence of 1.5 wt% of DCP (red ring symbols) and PLA/GTR_c_ 15% in presence of 1.5 wt% of DCP (blue diamond symbols). All blends were processed at 170 °C in presence of Irganox (0.2 wt% of total amount of material).

**Figure 6 polymers-13-01496-f006:**
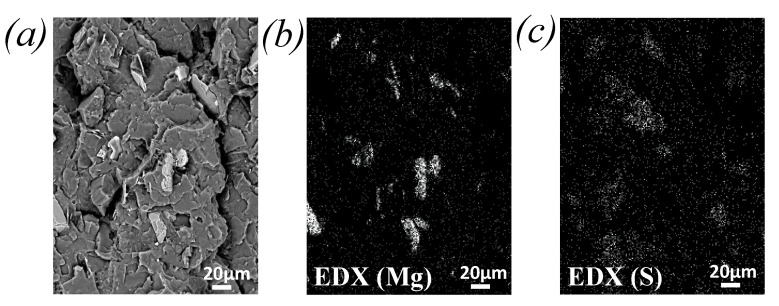
(**a**) SEM images at a magnitude 300× of fracture surface of PLA/GTR_a_ 15 wt% sieved 230’s mesh and obtained after uniaxial tensile test up to failure, (**b**) mapping of the magnesium element (Mg) dispersed into the PLA matrix obtained from chemical analysis by EDX and (**c**) mapping of the Sulphur element (S) indicative of the GTR location obtained from chemical analysis by EDX.

**Figure 7 polymers-13-01496-f007:**
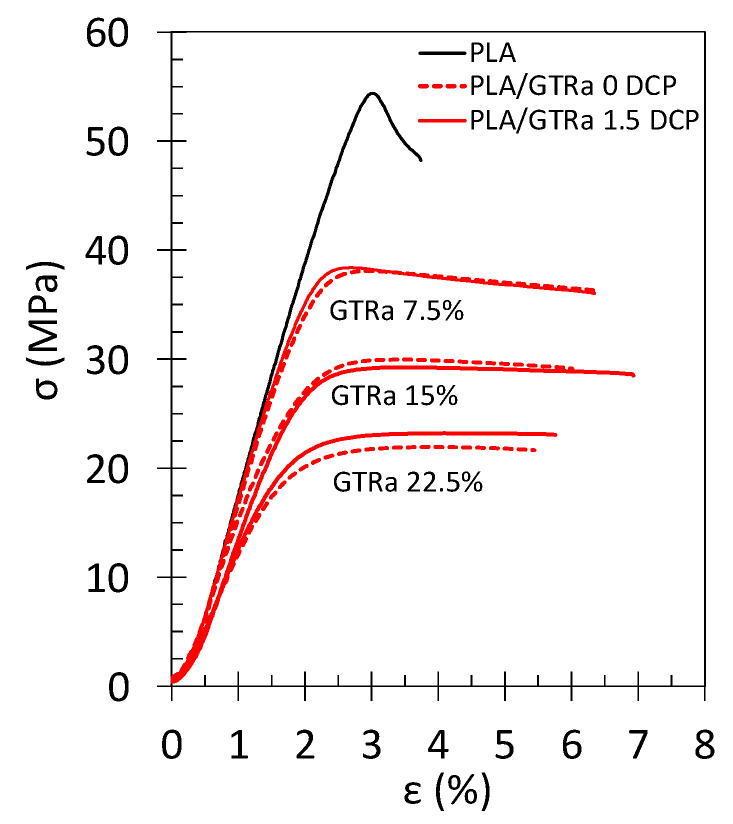
Engineering stress–strain curves of tensile test performed at 10 mm·min^−1^ and at room temperature on the neat PLA without DCP (black continuous line), PLA/GTR_a_ 7.5, 15 and 22.5% sieved 120’s mesh without DCP (red dotted lines) and PLA/GTR_a_ 7.5, 15 and 22.5% sieved 120’s mesh with 1.5 wt% DCP (red continuous line). All blends were processed at 170 °C in presence of Irganox (0.2 wt% of total amount of material).

**Figure 8 polymers-13-01496-f008:**
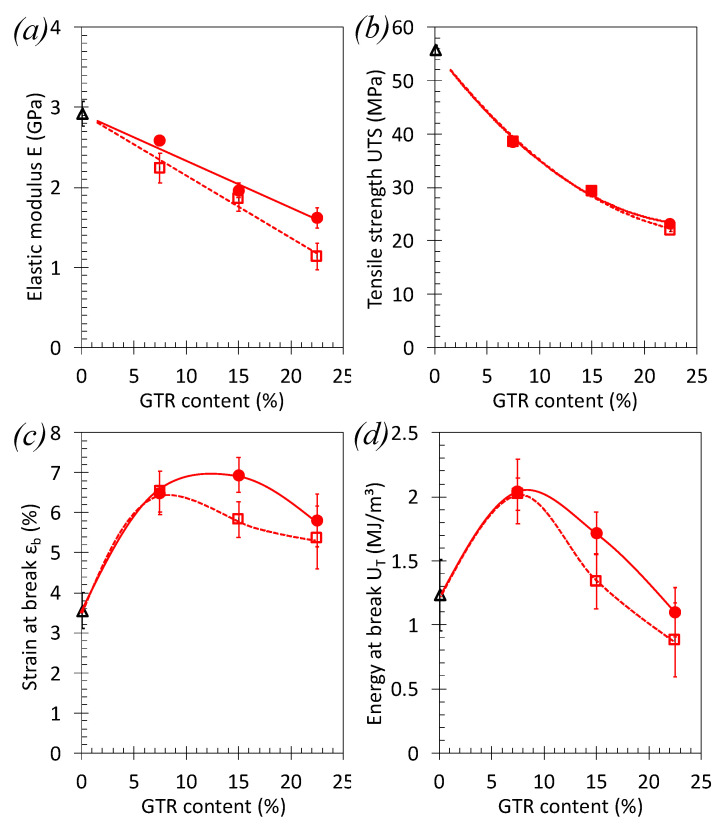
Effect of the GTR content on the tensile properties of PLA/GTR blends. (**a**) elastic modulus, E_T_, (**b**) tensile strength UTS, (**c**) strain at break, ε_b_, and (**d**) energy at break, U_T_, from tensile test performed at 10 mm·min^−1^ and at room temperature on the neat PLA in absence of DCP (black triangle symbols), PLA/GTR_a_ in absence of DCP (red unfilled square symbols) and in presence of 1.5 wt% of DCP (red filled circle symbols). All blends were processed at 170 °C in presence of Irganox (0.2 wt% of total amount of material). The continuous and dotted lines are guide for the eyes for the series of data associated with blend using DCP and without DCP, respectively.

**Figure 9 polymers-13-01496-f009:**
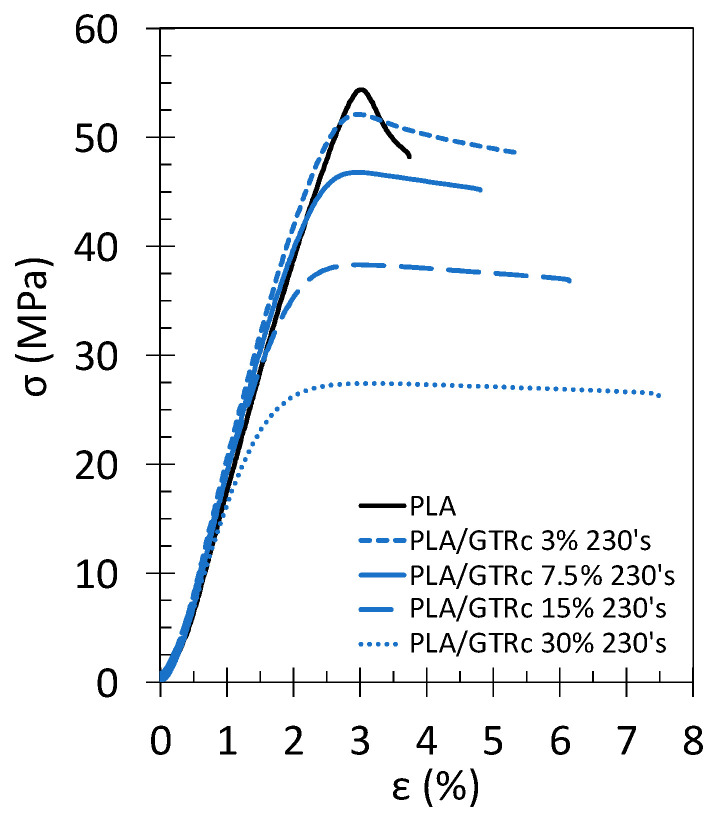
Engineering stress–strain curves of tensile test performed at 10 mm·min^−1^ and at room temperature on the neat PLA without DCP (black continuous line), PLA/GTR_a_ 7.5, 15 and 30 wt% sieved 230’s mesh with 1.5 wt% DCP (blue lines). All blends were processed at 170 °C in presence of Irganox (0.2 wt% of total amount of material).

**Figure 10 polymers-13-01496-f010:**
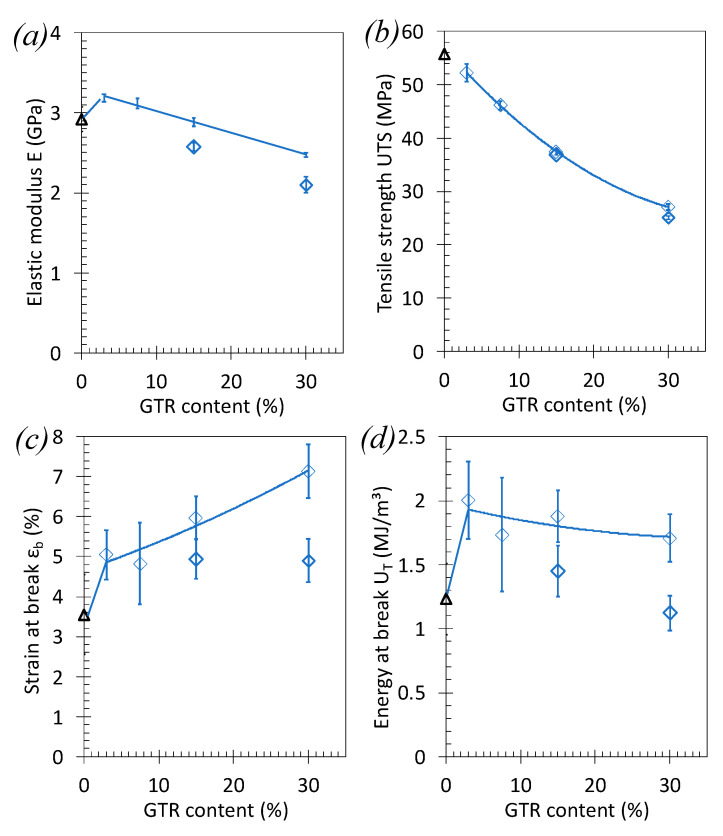
Effect of the GTR content on the tensile properties of PLA/GTR blends. (**a**) Elastic modulus, E_T_, (**b**) yield strength σ_y_, (**c**) strain at break, ε_b_, and (**d**) energy at break, U_T_, from tensile test performed at 10 mm. and at room temperature on the neat PLA (black triangle symbols), PLA/GTR_c_ in presence of 1.5 wt% of DCP (blue diamond symbols). All blends were processed at 170 °C in presence of Irganox (0.2 wt% of total amount of material).

**Figure 11 polymers-13-01496-f011:**
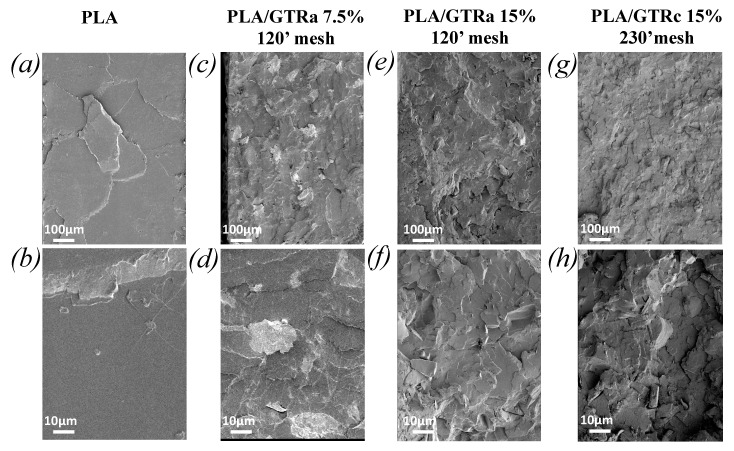
SEM images of fracture surface at two magnitudes x 100 (top) and x 1000 (bottom) of neat PLA (**a**,**b**), PLA/GTR_a_ 7.5 wt% with DCP (**c**,**d**), PLA/GTR_a_ 15 wt% with DCP (**e,f**), PLA/GTR_c_ 15 wt% with DCP (**g**,**h**).

**Figure 12 polymers-13-01496-f012:**
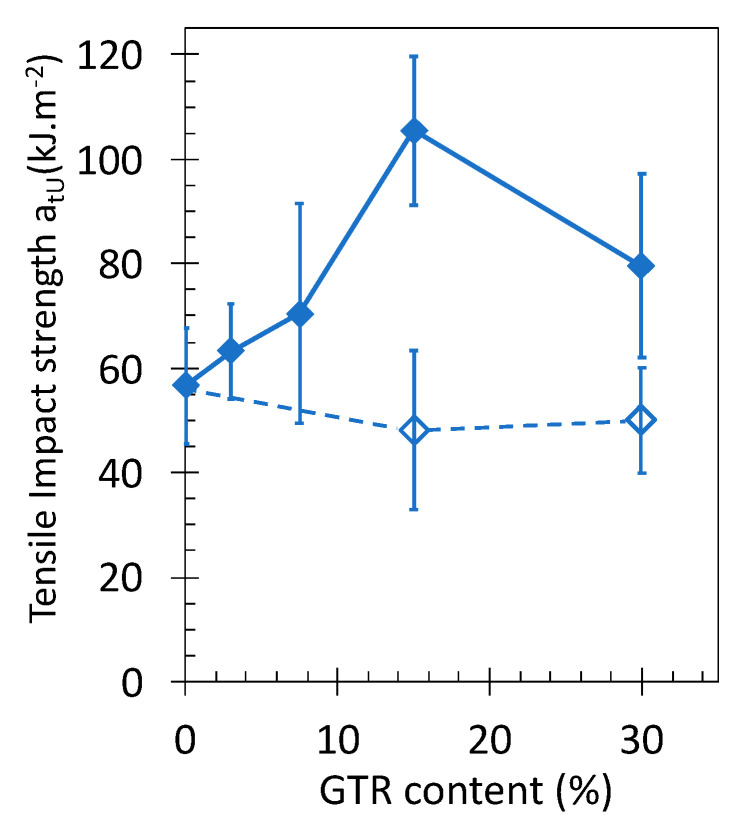
Tensile impact strength of neat PLA and PLA/GTR_c_ blends in presence of 1.5 wt% of DCP (filled diamond symbols) and without DCP (unfilled diamond symbols). All blends were processed at 170 °C in presence of Irganox (0.2 wt% of total amount of material). The continuous and dotted lines are guide for the eyes for the series of data associated with blend using DCP and without DCP, respectively.

**Table 1 polymers-13-01496-t001:** Code of the processed blends using various GTR grinding processes, sieving meshes and PLA/GTR quantities. All blends were prepared with 0.2 wt% Irganox^®^ 1010. GTR_a_ states for ambient grinded GTR and GTR_c_ states for cryo-grinded GTR.

Sample Code	GTR Weight Content (%)	DCP (per 100 g of GTR)	Pre-Treatment of the GTR Powder	Particles Mesh Size (µm)
PLA	0	-	-	-
PLA/GTR_a_ 15% Y ’s	15	0	Ambient grinding	Y = 40 ’s; 80 ’s; 120 ’s
PLA/GTR_a_ 15% Y ’s	15	1.5	Ambient grinding	Y = 40 ’s; 80 ’s; 120 ’s
PLA/GTR_c_ 15% Y ’s	15	1.5	Cryogenic grinding	Y = 120 ’s; 230 ’s
PLA/GTR_a_ X% 120 ’s	X = 7.5; 15; 22.5	0	Ambient grinding	120 ’s
PLA/GTR_a_ X% 120 ’s	X = 7.5; 15; 22.5	1.5	Ambient grinding	120 ’s
PLA/GTR_c_ X% 120 ’s	X = 3; 7.5; 15; 30	1.5	Cryogenic grinding	120 ’s
PLA/GTR_c_ 15% 120 ’s	X = 15	0	Cryogenic grinding	120 ’s

## Data Availability

The study did not report any data.

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
