# Peer review of "Poly (Lactic Acid)/Ground Tire Rubber Blends Using Peroxide Vulcanization"

_polymers, 2021, doi:10.3390/polym13091496_

Round 1

Reviewer 1 Report

The work deals with preparing PLA composites using GTR, which is interesting and meaningful. However, the authors should clarify the following questions before acceptance.

 Comments:

1. There are a few grammatical errors, such as “the finest sieved GTR contain the largest the amount…”. It can be corrected as “the finest sieved GTR contains the largest amount…

2. No evidence can prove that clay particles were dispersed into the PLA matrix. Maybe the authors can observe the dispersion of the particles on the cryo-fractured surface of the composites by liquid notrogen.

3. It is well known that PLA is a biodegradable thermal plastic, which is its superiority. However, addition of non-biodegradable GTR will weaken this superiority. Please clarify this issue.

4. Please describe the potential application of PLA/GTR composites in introduction part.

5. Why was the mass of dry sample weigh after swelling rather than before swelling?

6. Why was the samples for impact test not notched?

7. “The remaining mass above 500 °C is ascribed to non-rubber components…”. Was not the rubber carbonized?

8. How to get the size of the particles?

9. I suggest the authors to choose a proper solvent to extract PLA matrix of the composites and separate the grafted GTR particles, then characterize the grafting degree, which helps to understand the grafting process and mechanism.

Reviewer 2 Report

The manuscript presents some interesting results on the blend of Poly (Lactic Acid) and Ground Tire Rubber.  However, I have some concerns on this contribution:

  1. No need for the abbreviations like PLA in the title
  2. The abstract is not clear, the objectives, methodology and the results are not clear.
  3. The abstract should be rewritten.
  4. What is the motivation for blending biopolymer thermoplastic polymer with thermoset material?
  5. What are the applications of the produced composite?
  6. The introduction contains many abbreviations without definition.
  7. The molecular weight of PLA should be given.
  8. In Table 1; what is the difference between the composite in the 2nd raw and the composite in 3rd raw?
  9. FTIR analysis is necessary for this study, to determine if there is a chemical interaction between PLA and GTR and to identify the role of DCP.
  10. The EDX analysis of the elements for different types of GTR should be provided in addition to the SEM to support the results.
  11. The dot and solid lines in Fig. 7 should be defined.
  12. Actually, the effect of DCP on tensile properties is within the error limits. This point should be discussed.
  13. Error bars are missing in Fig. 9a and 9b.
  14. Using one point (15 wt.%) as a reference material is not enough to show the effect of DCP, like what is presented in Fig. 9 and 11.
  15. The properties of the developed composites should be compared with other PLA based composites.
  16. More recent references should be cited.

Reviewer 3 Report

Dear Authors,

The proposed research represents a very promising concept for the reuse of vulcanizate from end-of-life tires (ELTs). The approach taken by the authors allows for efficient integration of the vulcanizate into the organic matrix by chemically bonding it (despite the relative chemical inertness and heterogeneity of the resin). It is the mechanical properties of the PLA/GTR composite that are most important in the context of this study. An important aspect of the research is the significance of the GTR particle size for obtaining composites with the best properties. An increase in the concentration of GTR particles above a certain point leads to a deterioration in the mechanical properties of the resulting composite.  Overall, this study extends the knowledge on the possibilities of recycling ELTs in order to obtain final materials suitable for practical use. 
